# What are the spatio-temporal differentiation characteristics and driving factors of the coupling coordination degree between green finance and ecological efficiency? Evidence from 84 cities in western China

Dalai Ma[1], Bitan An [1]*, Zuman Guo[1], Jiawei Zhang[1], Fengtai Zhang[1,2,3], Ruonan Chang[4], Yin Yan[1]

**1** School of Management, Chongqing University of Technology, Chongqing, China, **2** Institute for Ecology and Environmental Resources, Chongqing Academy of Social Sciences, Chongqing, China, **3** Research Center for Ecological Security and Green Development, Chongqing Academy of Social Sciences, Chongqing, China, **4** School of Economy and finance, Chongqing University of Technology, Chongqing, China

* 13031953239@stu.cqut.edu.cn

## Abstract

Facilitating the coordinated and effective progress of green finance (GF) and ecological efficiency (EE) stands as a potent approach to support our nation in attaining sustainable development goals. This paper Utilized panel data encompassing 84 cities in Western China spanning from 2007 to 2021, this paper empirically analyzes the spatio-temporal characteristics and driving factors of the coupling coordination degree between green finance and ecological efficiency (CCD-GFEE) in western cities. The findings indicate that: (1) The level of GF demonstrates a rising trajectory, with significant regional disparities. Besides, the high level area progressively moves from the northwest to the southwest. (2) On the whole, urban EE demonstrates a relatively elevated level, but it still fails to reach DEA effectiveness. Compared to the northwest region, the southwest region has poorer efficiency. (3) The CCD-GFEE in western China showing a slight growth trend. The coupling coordination degree (CCD) in Northwest China is higher than that in Southwest China, and cities with higher CCD are primarily found in Inner Mongolia, Sichuan Province and Shaanxi Province. Within them, the CCD of Chengdu is the highest, Chongqing has achieved the largest stage leap. (4) The global Moran's I consistently remained positive and exhibited a tendency of initially rising and subsequently falling, indicating that the spatial aggregation effect of CCD-GFEE first increased and then decreased. (5) The CCD-GFEE driving factors are examined using the spatial econometric model, and it has been observed that the impact of population size and government intervention on CCD-GFEE is negative, while the impact of industrial structure, technological progress and economic level on the coupling and coordination of CCD-GFEE is positive.

**Data availability statement:** All relevant data are within the paper and its Supporting Information files.

**Funding:** Research on uncoordinated coupling identification and coordinated development path of green finance and ecological resources in western China" (No. 21CJY033). The funders had no role in study design, data collection and analysis, decision to publish, or preparation of the manuscript.

**Competing interests:** The authors have declared that no competing interests exist.

Finally, the paper presents certain policy enlightenments to guide the coordinated development of GF and EE from the aspects of GF system formulation, economic construction and technological progress.

## 1. Introduction

Since the Industrial Revolution, The swift growth of the global economy has also given rise to numerous environmental issues such as excessive energy consumption and massive emission of pollutants. The severe ecological problems caused by the resulting resource shortage and global warming have seriously jeopardized the life safety of people of all ethnic groups in the world. It has evolved into a shared concern for nations worldwide. Finance, as the center of the contemporary economy, achieving the harmonious development of the economy and ecology necessitates financial innovation [1]. Traditional financial by making polluters get low-cost loans, promote the industrialization and increase carbon emissions, to accelerate the deterioration of the quality of the environment [2]. In this context, as a concept derived from the theory of green development in practice, GF has been widely concerned by the academic community. Unlike traditional finance, which focuses solely on maximizing economic benefits, Green finance seeks to streamline the funding of sustainable development initiatives, particularly those focused on addressing climate variability and environmental issues [3]. China is the first country to formulate a comprehensive framework for promoting GF. A number of Chinese government departments have jointly issued policies and regulations such as the "Guiding Principles for Establishing a Sustainable Financial System" and the "Recommendations for Enforcing Environmental Protection Policies and Regulations to Mitigate Credit Risks", confirming the important position of green finance in ecological and environmental protection [4].

Twelve provinces, autonomous areas, and municipalities, including Chongqing, Sichuan, and Gansu, make up western China. There are abundant resource deposits in the western region. For a long time, in addition to safeguarding the natural environment, the western area of China must supply raw materials and energy to the nation's east and central regions [5]. Since the Western Development Policy came into effect, unsustainable urban planning and excessive urban land expansion have resulted in a number of socioeconomic and environmental issues, including the destruction of ecosystems, the tension between the scarcity of urbanization resources and the growth of urban population has become increasingly prominent [6]. As resource-intensive development often leads to environmental damage, it is becoming increasingly important to reconcile urban economic growth with resource capacity [7,8]. Therefore, promoting the Western Region's urban economy and ecological to work together has taken precedence [9]. From the macro level, green finance can promote the development of green industry and optimize the balance of ecological environment by giving priority to green industry policies. In addition, green finance provides financial support for ecological and environmental protection through green securities and green investment; Finally, when the government solves environmental problems, the specific measures related to financial instruments such as fiscal funds and bank

loans have promoted ecological and environmental protection. At present, green finance in western China started late, the development is relatively backward, the relevant policies are not perfect, and the ecological environment in western China has not achieved coordinated development.To better study the CCD of urban GF and EE in western China, it is necessary to apply the system view and collaborative thinking. The spatiotemporal evolution of the two systems and their coupled coordination level will be examined in this research, explore their driving factors, and propose targeted policies to encourage the two systems' development in tandem and in unison. Based on the above background, this study intends to address the following core issues: What is the current status of the coupling and coordination between green finance and the ecological environment system in western cities? What are the characteristics of spatio-temporal differentiation? What are the key factors driving the coupled and coordinated development of the two systems? How can the level of coordinated development of green finance and ecological environment in western cities be enhanced through institutional innovation and policy optimization? To better study the coordinated development of green finance and ecological efficiency in the western region of China, it is necessary to apply the systems view and collaborative thinking. This paper will construct the index system of the two systems, measure their coupling coordination level by using the coupling coordination degree model, and conduct spatio-temporal evolution analysis by using methods such as spatial econometric analysis. Finally, the spatial econometric model will be used to explore its influencing factors. To promote the coupled and coordinated development of the two systems, targeted policies are proposed.

## 2. Literature review

### 2.1. Green Finance

In response to the escalating global demand for increased corporate involvement in environmental sustainability, financial institutions are realizing more and more how important it is to change their business strategies to prioritize environmental responsibility [4]. Based on this understanding, the concept of GF is put forward. However, because international financial organizations, scholars and governments have defined it according to their own understanding and practice, there's no single accepted definition for this concept. In the early days, scholars generally endowed GF with the function of leverage for sustainable growth in both the economy and society, believing that it played an effective financial support role in environmental protection and encouraged the harmonization of environmental preservation and economic development [10]. Later, scholars have different views on green finance. Some scholars define green finance as environmental finance, whose main role is to address the issues of resource scarcity and pollution in the environment [11]. Zhao, Li [4] sees GF as a financing tool to aid in efforts for sustainable development and environmental preservation. Despite the fact that experts interpret the meaning of "green finance" differently, in general, they have reached a consensus that GF is mainly used to address environmental issues and provide financial support for environment-related economic activities, in order to achieve the economy's and the environment's integrated development.

GF facilitates the development of environmental protection projects by developing and utilizing GF tools, thus encouraging the modification of the industrial structure to environmental friendliness. So far, academics cannot agree upon metrics to assess the state of GF development. Cao, Zeng [12] measures China's overall development in GF using green credit. Peng, Wang [13] determined the level of GF by taking into account government support, green loans, green insurance, and green investments. To gauge the level of GF in China's Yangtze River Delta, Xie, Ouyang [14] used green investment and green credit.

### 2.2. Ecological efficiency

The original concept of EE comes from ecology, which speaks to the ecosystem's effectiveness in using its resources while providing ecological services. The concept of eco-efficiency was first proposed by Schaltegger, S. [15] and promoted worldwide by the Organization for Economic Cooperation and Development (OECD) and the World Business Council for Sustainable Development (WBCSD), according to which EE is the ratio of economic input to environmental

costs, and this definition is widely recognized by scholars [16]. EE is frequently regarded as a business idea, and some scholars have explored eco-efficiency in specific industries, such as transportation [17], Dairy industry [18], Logistics industry [19] et al. In addition, scholars in the past also have learned from the global, national [20,21], region [22,23], province [24–26]and city [27]. The commonly used method to measure EE is the comprehensive evaluation method. Among them, the main methods to construct index system for comprehensive evaluation include material flow method [28], ecosystem service value accounting method [29], factor analysis method, and life cycle evaluation method [30], Energy Analysis [31], Data Envelopment Analysis (DEA) and its expanded model [32,33] et al. EE is a measure of the comprehensive efficiency of ecological surroundings and economic system, considering the environmental cost of economic growth, and can objectively reflect the level of sustainable economic development. EE can be understood as an efficiency value, so scholars tend to prefer DEA and its extended model when measuring eco-efficiency. For instance, Wang, Wu [34] calculated the China Industrial Sector's 37 sub-industries' EE and Industrial Upgrading Index (IUI) utilizing the super-efficiency slack-based measure (SESBM) model and the comprehensive weighting approach based on subjective and objective weights. Xiao, Wang [35] presented a two-stage network DEA framework integrating the segments of the government and industry and assessed the EE of 84 resource-based communities in the years following the financial crisis.

### 2.3. Coupling of green finance and ecological environment

The study of the harmonious development relationship between national economy and environment has constantly centered on the economy's sustainable development [36]. A few academics have studied the link between GF and environment. For example, in terms of the relationship between GF and energy, Lee, Wang [37] found that GF contributes significantly to energy efficiency. According to Zhang, Hao [38], the creation of GF can successfully reduce energy poverty over the long run. In terms of carbon emission reduction, Ran and Zhang [39] found that Carbon emission reduction can be greatly aided by GF, and the effect of reducing carbon emissions of green finance is more significant in developed regions and western regions. Ren, Shao [40] found that GF can reduce carbon emissions. In addition, Lan, Wei [41] also found that the relationship between GF and industry-driven emissions has regional differences in the three categories of pollutants.

However, research on the connection between GF and the ecological environment is scarce. First of all, the execution of the reform in GF can encourage the significant enhancement of green innovation of enterprises, thus promoting the improvement of ecological environment [42]. Secondly, environmental sustainability benefits from GF by improving the financial performance of environmentally conscious businesses [43]. According to Huang, Mbanyele [44], GF greatly enhanced high-polluting sectors' corporate environmental responsibility performance, thus combining green finance with economic benefits. Zhao, He [45] found that GF can amplify the benefits of industry structural reforms for the advancement of renewable energy. GF is also an important tool for enhancing the environment and halting global warming [46]. In turn, the message bearers in the GF system are the environment, renewable energy, and water markets [46]. In turn, a high degree of EE will encourage the growth of economic activity in a sustainable manner and promote the development of GF [3,47]. In addition, the use of renewable energy promotes the development of GF [48]. The above studies show that there is a mutual influence between GF and ecological environment, but there are few studies on the interaction between the two quantitatively.

There are many methods used by the academic circle to test the way two systems interact, mainly including: system dynamics model [49], machine learning model [50], Vector autoregressive (VAR) model [51,52], support vector machine model [53], Coupling coordination model (CCDM) [54,55] et al. The CCDM has gained the most traction because it allows for a quantitative evaluation of the degree of system cooperation to emphasizing the relationships between subsystems [4]. Therefore, using a CCDM, this research investigates the relationship between GF and EE in western China and the coupling and coordinated development.

In addition, Some scholars also based on exploratory spatial data analysis (ESDA) model [56], graphical detector model [56],Geographic detector model (OPGD) [57], spatial metrology model [58], Geographical weighted regression (GWR) model [59] explored the driving factors of GF and ecological environment. It mainly includes industrial structure [60], economic development level, environmental regulation [60], population structure [61], technological innovation [57], government support [62], urbanization level [56], energy intensity [56]et al.

It can be seen from the above literature review that most scholars use just a few indicators to measure GF, which lacks a certain comprehensiveness. In light of ecological efficiency and GF, numerous academics have carried out in-depth research respectively, but the research on the combination of GF and EE is relatively scarce. For the existing literature discussing the relationship between the two, the main research is the one-way influence process of GF on ecological efficiency, ignoring the interaction between them. Moreover, the current research only stays in the judgment of the status quo of the two systems, and does not further examine the driving elements that are fostering the two systems' coordinated development. Secondly, the research scope mainly focuses on regions and provinces, and there are few studies on cities. This paper's innovation resides in: (1) On the basis of previous researches, it builds an extensive GF index system from five aspects to comprehensively evaluate the development level of urban green finance in western China; (2) It explores the CCD relationship between urban GF and EE in western China, and few literatures have studied this issue, even less from the city level. (3) The interactive relationship between GF and EE is expanded. The CCDM was used to quantitatively analyze the CCD-GFEE in western China, and the spatio-temporal evolution was visualized by Argis software.

## 3. Data and methodology

### 3.1. Research area

There are twelve provinces and autonomous areas in Western China. including six provinces and municipalities in southwest China (Chongqing, Sichuan, Guizhou, Yunnan, Tibet and Guangxi) and six provinces and autonomous regions in northwest China (Shaanxi, Gansu, Qinghai, Xinjiang, Ningxia and Inner Mongolia). It should be noted here that due to the influence of data collection channels, in the key data sources, namely the "China Urban Statistical Yearbook (2007-2021)" and the "China Urban Construction Statistical Yearbook (2007-2021)", the data for many indicators of cities such as Lhasa, Xigaze, Chamdo, and Nagqu in Tibet, Haidong City and Haibei Tibetan Autonomous Prefecture in Qinghai, and Hami City and Altay in Xinjiang are severely lacking. Based on the principles of data integrity and operability, these cities with severely missing data were excluded from the evaluation in this study. Ultimately, 84 cities with relatively complete and representative data were selected as the research samples, accounting for over 60% of all cities in the western region and covering the majority of cities in the western region.

### 3.2. Sample Selection

The first green finance policy jointly released in 2007, which was titled Opinions on Implementing Environmental Protection Policies and Regulations to Prevent Credit Risks, is commonly recognized as the start of China's all-encompassing GF approach. Therefore, we set the starting point of the study as 2007 and selected 84 cities in western China as research samples.

### 3.3. Methods and models

The purpose of this study is to look into the development status of CCD-GFEE in cities in western China. To explore the allocation of CCD in different cities to promote the coordinated development of urban GF and EE in western China. Fig 1 displays the study's research design and methods. Firstly, for GF, we constructed GF indicators from five aspects. For EE, we measure EE from an input and output perspective based on the Super-EBM model. Secondly, the CCD-GFEE is computed using the CCDM, and its spatio-temporal evolution properties are examined. Finally, the SEM is used to analyze the driving factors of CCD-GFEE.

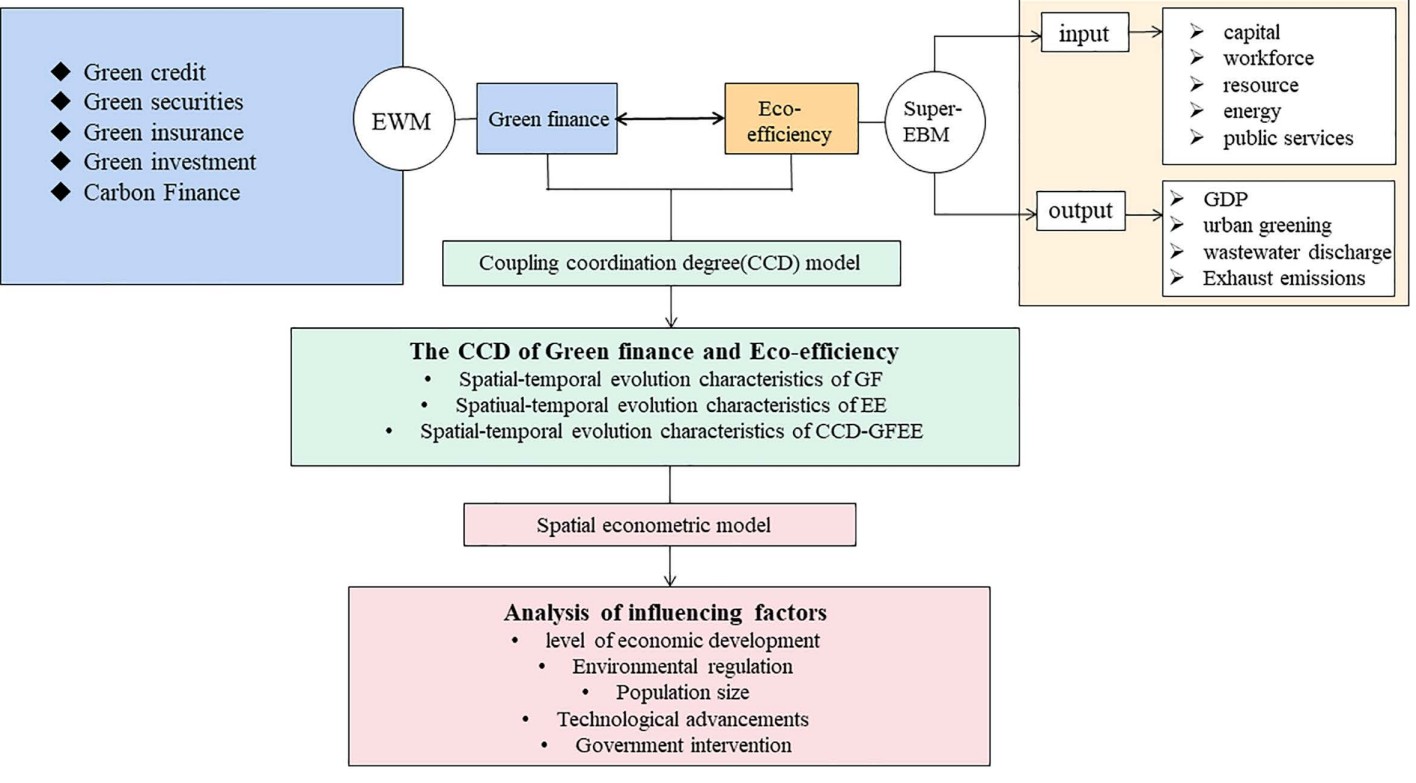

**Fig 1. Conceptual model.**

**3.3.1. Entropy weight method (EWM).** EWM converts each index into entropy value through entropy value theory, and then evaluates the performance of the system comprehensively according to the weight distribution. Since the EWM involves the process of computation by taking the natural logarithm in the standardization process, in order to avoid the meaningless allocation of numbers, the computation formula for processing the positive and negative indices is as follows [63].

(1) Data standardization.

$$Mij = \frac{X_{ij} - \min(X_{ij})}{\max(X_{ij}) - \min(X_{ij})} + 0.001 \tag{1}$$

$$Mij = \frac{\max(X_{ij}) - X_{ij}}{\max(X_{ij}) - \min(X_{ij})} + 0.001 \tag{2}$$

Where $Mij$ is the normalized matrix, $X_{ij}$ is the initial data, The original data's maximum and minimum values are denoted by $\max(X_{ij})$ and $\min(X_{ij})$, respectively.

(2) Calculate the proportion of the i-th object under the j-th index $Q_{ij}$:

$$Q_{ij} = M_{ij} / \sum_{j=1}^{n} M_{ij} \tag{3}$$

(3) Calculate the entropy $e_j$ of the j-th index, N is the total number of samples in this case:

$$e_j = -k \sum_{j=1}^{n} Q_{ij} \times \ln\left(Q_{ij}\right), k = 1/\ln(N)$$

(4)

(4) Calculated the difference coefficient $c_j$:

$$c_j = 1 - e_j$$

(5)

(5) Calculated weight $c_j$:

$$w_j = c_j / \sum_{j=1}^{n} c_j$$

(6)

(6) Calculate composite score $S$:

$$S = \sum_{j=1}^{n} w_{ij} \times M_{ij}, \ \ and \sum_{j=1}^{n} w_{ij} = 1$$

(7)

Within the formula, $S$ is the evaluation result of the comprehensive level of GF. The weight matrix $w_{ij}$ is determined using the entropy weight method.

**3.3.2. Super-EBM model.** The conventional DEA model does not take into account the relaxation factors and is unable to compare efficient DMU with a value 1.In addition, Its measurement results are simply radial and angle, with limited accuracy, and it is unable to address the issue of environmental pollutants as an undesirable output [64]. A hybrid distance function model containing both radial and non-radial elements was proposed by Tone and Tsutsui [65], Epsilon-based Measure (EBM) model. But when there are multiple effective DMU (that is, efficiency values are all 1), the EBM model cannot further compare efficiency values. Super-EBM is a DEA method that takes relaxation variables and unexpected output into account, it achieves the efficient fusion of non-radial and radial, and it guarantees that the optimal solution is in the dimensionless state while raising the measurement value of the effective DMU to > 1 [66], can solve the problem of further ranking of effective elements, and utilised extensively for ecological efficiency, energy efficiency and innovation efficiency [67]. Here is the formula for Super-EBM:

$$\rho^* = \min \frac{\theta - \varepsilon_x \sum_{i=1}^{m} \frac{\omega_i^- s_i^-}{x_{ik}}}{\varphi + \varepsilon_y \sum_{r=1}^{s} \frac{\omega_r^+ s_r^+}{y_{rk}} + \varepsilon_b \sum_{p=1}^{q} \frac{\omega_p^{b-} s_p^{b-}}{b_{pk}}}$$

$$\begin{cases} \sum_{j=1}^{n} \lambda_j x_{ij} + s_i^- = \theta x_0 \quad i = 1, 2, \cdots, m \\ \sum_{j=1}^{n} \lambda_j y_{rj} + s_r^+ = \varphi y_{rk} \quad r = 1, 2, \cdots, s \\ \sum_{p=1}^{n} \lambda_j b_{pj} + s_p^{b-} = \varphi b_{pk} p = 1, 2, \cdots, q \\ \lambda_j \cdots 0, s_r^+ \cdots 0, s_i^- \cdots 0, s_p^{b-} \cdots 0 \end{cases}$$

(8)

Within the formula above, $\rho^*$ is the objective function value; $\varepsilon_y$ and $\varepsilon_b$ are the important parameters; The aforementioned items' respective weight indicatiors are denoted by $\omega_r^+$ and $\omega_p^{b-}$; The relaxation variable for class r's anticipated output is $s_r^+$; $\varphi$ is the output expansion ratio; $s_p^{b-}$ is a relaxation variable with an undesired output of class p; The decision unit j's type p unwanted output is $b_{pj}$; The decision unit k's p-class undesirable output is $b_{pk}$; The quantity of anticipated outputs is q.

### 3.3.3. Coupling coordination degree(CCD) model.

CCD model is established to calculate CCD-GFEE. Taking into account the different measurement methods used for GF and EE, the data is standardized and then calculated by plugging it into the formula [60].

$$C = 2 * \sqrt{\frac{S_1 * S_2}{(S_1 + S_2)^2}} \tag{9}$$

$$T = a \times S_1 + b \times S_2 \tag{10}$$

$$D = \sqrt{C \times T} \tag{11}$$

Among them, $S_1$ and $S_2$ represent standardized indicators of GF and EE, respectively. D represents the CCD, the coupling degree is denoted by C, and T stands for the comprehensive level of EE and GF; Undetermined coefficients are denoted by a and b. Because EE and GF are deemed equally significant in this paper, a=b=0.5 is used.

### 3.3.4. Spatial autocorrelation model.

Finding out if connected systems are correlated based on geographic factors can be done using spatial autocorrelation [68]. We tested the geographic correlation of CCD-GFEE using the Global Moran'l in order to understand its spatial heterogeneity. The following is the computation formula:

$$Moran's\mathrm{I} = \frac{n\sum\limits_{i=1}^{n}\sum\limits_{j=1}^{n}Wij(xi-\bar{x})(xj-\bar{x})}{\sum\limits_{i=1}^{n}\sum\limits_{j=1}^{n}Wij\sum\limits_{i=1}^{n}(xi-\bar{x})^2}$$

$$Wij = \begin{cases} \frac{1}{hij}, i \neq j \\ 0, i = j \end{cases} \tag{12}$$

Where, $n$ is the number of cities, $x_i$ and $x_j$ represent CCD-GFEE of cities $i$ and $j$ respectively, $\bar{x}$ is the CCD-GFEE average value. The spatial weight matrix is denoted by $W_{ij}$, the geographic separation between two cities is denoted by $d_{ij}$. In this work, the geographical distance matrix is constructed according to the city's latitude and longitude, which can better reflect the spatial correlation of discrete spatial units [69].

### 3.3.5. Spatial econometric model.

To explore the system's driving forces, a spatial econometric model can be used if the distributed distribution of CCD according to geographic characteristics is shown to be spatially correlated [60]. The spatial dependence issue which cannot be solved by linear regression analysis is explicable by the spatial econometric model [70]. Within this research, spatial autocorrelation model (SAR) and spatial error model (SEM) were used to analyze the drivers of CCD-GFEE.

SAR emphasizes that the explained variables in a certain spatial range are influenced by neighboring spatial units in addition to the explanatory variables [71], the following is an expression for the formula:

$$y = \rho Wy + X\beta + \epsilon \tag{13}$$

Where $y$ represents the explained variable, the total number of cities is represented by $n$, the total number of explanatory variables is denoted by $k$. $X$ represents a matrix of $n*k$, stands for explanatory variable, $\beta$ is the $k*1$ vector for the

explanation variable's coefficient, the correlation coefficient for spatial regression is denoted by ρ , The spatial weight matrix is represented by $W$, the spatial lag term of the explained variable is denoted by $W_y$, ε represents a random error term vector.

SEM adds a spatially dependent error term to solve the spatial dependence problem according to the function of the spatial error term [72]. The formula can be written as follows:

$$\begin{cases} y = X\beta + \epsilon \\ \epsilon = \lambda W\epsilon + \mu \end{cases}$$

(14)

Where, spatial autocorrelation error term is denoted by $W_\epsilon$, the coefficient of spatial inaccuracy is represented by $\lambda$, μ is the error vector that is random of the positive distribution, and the other variables have the same meaning as (13).

### 3.4. Index system construction

Based on the state of GF and EE development in China and following the principles of data availability, scientific and systematic, a set of indicators was created for this study to characterize GF and EE.

**3.4.1. Green finance index system.** Numerous academics have created a range of assessment metrics to assess the level of GF. This work examines GF from five different acpets: green investment, green securities, green credit, green insurance, and carbon finance [73,74]. The specific index system is shown in Fig 2.

Green credit: In the province's entire credit portfolio, green credit is defined as the percentage of total credit allotted to environmental initiatives,and it shows how much support financial institutions give to funding environmental projects [75]. We chose the annual green credit balance of 34 listed banks as the base sum. Since there is no data available for cities at the prefecture level, we weighted the green credit balance by the proportion of financial institutions' outstanding loans in each city at the prefecture level and the banking institutions' loan balance in the whole country to obtain the green credit balance data of each prefecture-level city.

Green securities: Regarding green securities, we draw on the practice of Zhang, Zhu [73], and mainly select the percentage of six highly energy-intensive companies' A-stock market value and A-share market value of energy conservation and environmental protection industry to measure the development degree of green securities(The six industries with high energy consumption include chemical raw materials and chemical products manufacturing industry, non-metallic mineral products industry, petroleum, coal and other fuel processing industry, electric power, heat and gas production and supply industry, the industry of rolling and smelting ferrous metal, the non-ferrous metal rolling and smelting sector.).

Green investment: The development of GF is significantly aided by government fiscal expenditure [76]. The government has adopted macro-control and increased financial investment in environmental preservation, which will support the growth of the industry for environmental protection. As for green investment, we use the percentage of spending on energy conservation and environmental protection and the fixed asset investment in the construction of municipal public facilities for landscaping to measure [73,75].

Green insurance: The market share of environmental liability insurance in insurance products and the insurance industry's concentration on environmental concerns are reflected in green insurance [75]. In 2013, China started to introduct corporate environmental liability insurance and long-term data are lacking. The agricultural insurance scale is therefore used to measure green insurance. [60].

Carbon finance: Based on each region's ratio of GDP to CO emissions, carbon financing indicates how developed the carbon trading markets are [60]. To better represent the level of funding allocated to low-carbon, we also added the proportion of carbon emissions in each region to the outstanding loans of financial institutions as a measure.

**3.4.2. Construction of ecological efficiency index system.** According to the Super-EBM model, this paper uses input indicators, expected output indicators and unexpected output indicators to measure urban EE. Table 1 displays the specific indicator system.

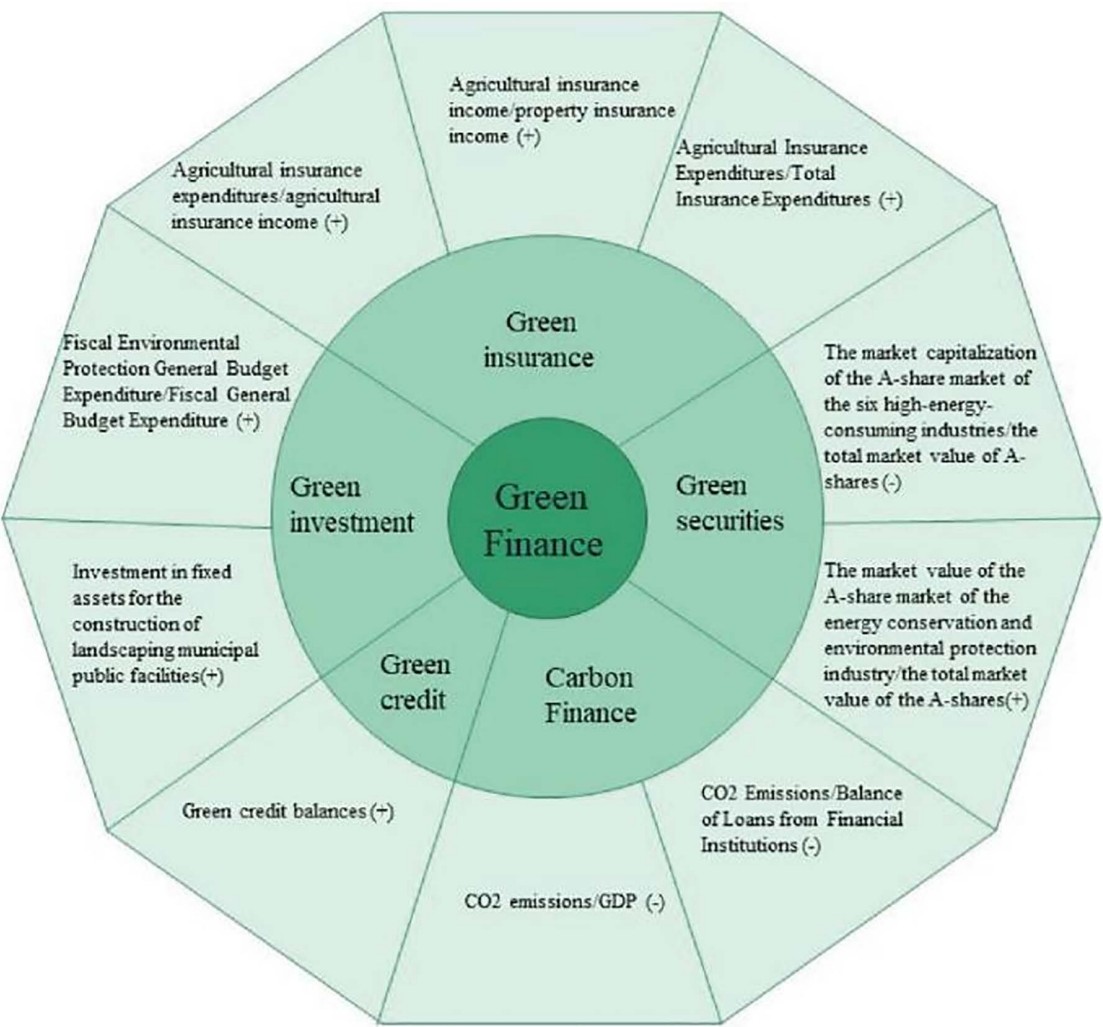

**Fig 2. Green finance index system.**

**Table 1. Ecological efficiency index system.**

| Indicator Type | Category | Indicator description | Unit |
|---|---|---|---|
| Input index | Capital | Total investment in fixed assets | 10 thousand yuan |
| | Labor force | Employment number | Person |
| | Resources | Total water supply | 10 thousand cubic meters |
| | | Use of liquefied petroleum gas | Ton |
| | Public service | Number of schools | Institute |
| | | quantity of beds in hospitals and health centers | Unit |
| Expected output | Urban economic development level | GDP | 100 million yuan |
| | Urban greening | Green land rate in built-up area | % |
| Undesirable output | Waste water Discharge | Industrial waste water discharge | 10 thousand tons |
| | Exhaust emission | Industrial sulfur dioxide emissions | Ton |

Input indicators include capital [5], labor force [5,16], resources [16], and public services.In this work, Capital is measured as the entire investment in fixed assets [16], and the total number of personnel in employment is used to measure labor force [16]. The total water supply and liquefied petroleum gas consumption are used to measure resource input [16]. In order to better represent urban eco-efficiency, public service input is added to the index system, where public service input is measured by the number of schools and the quantity of beds in hospitals and health centers.

Expected output is measured in two ways: the degree of urban greening and economic development [33]. Among them, the level of urban economic development is measured by GDP, and green land rate in built-up areas is used to represent urban greening.

Undesirable outputs are measured by effluent and exhaust emissions [16,33]. Among them, industrial waste water emissions represent waste water emissions, and emissions of industrial sulfur dioxide represent waste gas emissions

**3.4.3. Construction of CCD-GFEE driver factor index system.** According to the above, urban CCD-GFEE in western China is spatially and temporally different. Combined with the analysis of the driving mechanism of the coordinated development of GF and EE in the theoretical basis, this section adopts spatial econometric model and other methods to further explore the driving elements behind the two's coordinated development. Considering earlier research, this study mainly constructs the driving factor index system from six aspects: Economic level (EL) [60]、 environmental regulation (ER) [62], industrial structure (IS) [62], population size (PS) [77], Technical Progress (TP) [62], Government intervention (GI) [62], The explained variable is CCD-GFEE. In order to avoid the problem that the coefficients are too small to show, population density, GDP per capita, and science and technology expenditures are logarithms. See Table 2 for specific indicators.

### 3.5. Data sources

The data were mainly collected from China Urban Statistical Yearbook, China Urban Construction Statistical Yearbook, China Industrial Statistical Yearbook as well as provincial and municipal statistical yearbooks, WIND database, CSMAR database,A small part of missing data was completed by linear interpolation method.

## 4. Results and analysis

### 4.1. Temporal and spatial changes of GF

**4.1.1. Time evolution trend of GF.** As illustrated in Fig 3, in general,the average level of GF is mainly distributed between 0.03 and 0.1, showing a relatively low level on the whole. From the standpoint of the temporal evolution trend, the overall average level increased from 0.047 in 2007 to 0.087 in 2021, showing an upward trend. The growth rate of the southwest region is 93.5%, while that of the northwest region is 80.9%, indicating that the southwest regional growth is outpacing northwest regional growth.This may be due to the Southwest's advantageous physical location, the significance

**Table 2. CCD-GFEE driver index system.**

| Category | Index type | Index description | Unit |
|---|---|---|---|
| Interpretive index | EL | Per capita GDP | 10 thousand yuan |
| | ER | Urban sewage treatment rate | % |
| | IS | The proportion of tertiary industry | % |
| | PS | Population density | person/square kilometer |
| | TP | Spending on technology and science | 10 thousand yuan |
| | GI | Local fiscal expenditure/local GDP | % |
| Interpreted index | CCD-GFEE | | |

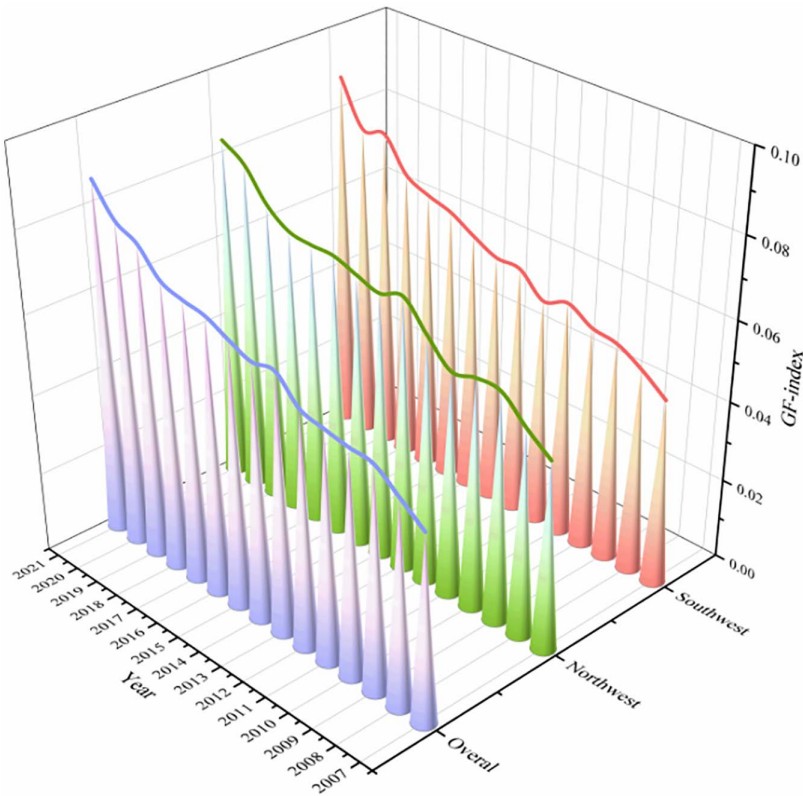

**Fig 3. Time evolution trend of green finance level.**

of progress in the economy and the strong awareness of environmental protection, so green finance shows a more rapid development.

**4.1.2. Spatial evolution trend of GF.** In this part, we analyze the results of the measurement of GF level, and study its temporal and spatial characteristics. For the purpose of more intuitively represent the spatial change characteristics of the GF level, the GF level was split into three tiers: low [0–0.04), medium [0.04–0.1) and high (>0.1). The GF level's spatial distribution from 2007 to 2021 is also described.

As illustrated in Fig 4, the overall level of urban GF development in western China is still at a relatively low level, with significant regional variations. In addition to the high level of GF development in some cities, the level of GF in other regions is always low, which indicates that most cities in western China have a large room for improvement. Among them, between 2007 and 2021, there were 15 high-level cities, up from just 2, and from 22 low-level cities in 2007–7 in 2021, the number of these cities declined, showing a notable improvement in the GF level in the western area.From Inner Mongolia in 2007 to Guangxi, Guizhou, Sichuan and other provinces in 2021, among them, the capital cities of these provinces are all high-level cities, which may be due to the superior geographic position of provincial cities, which pay more attention to the application of clean technologies while developing economy. Led to a higher level of GF. Among them, Inner Mongolia has the most high-level cities, and the GF level in Hulunbuir has always been at a high value, which may be owing to the superior geographical position of Hulunbuir, the capital city of Inner Mongolia, and greater focus on the common development of economy and environment, resulting in a higher level of green finance. Among them, Panzhihua and Zhaotong have always been at a low level. The economic development of these cities is lagging behind, the financial industry is

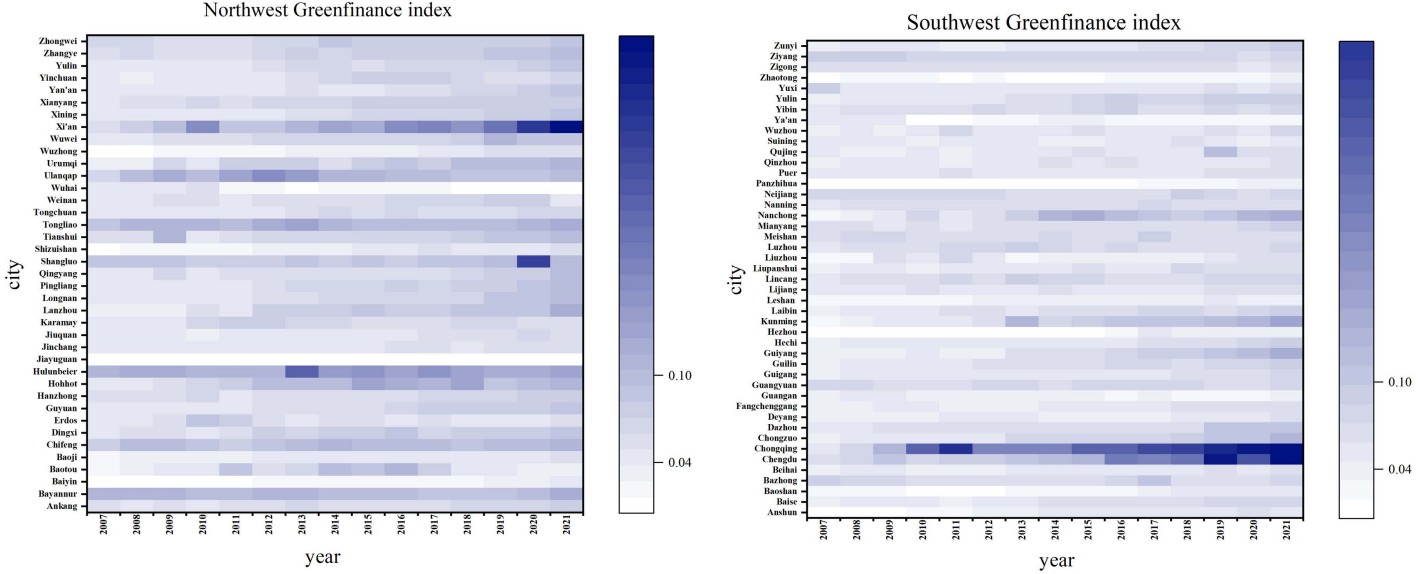

**Fig 4. Spatial evolution trend of green finance index.**

underdeveloped, and most of them take energy development as the economic pillar, and there is considerable environmental pollution, resulting in a low level of green finance.

### 4.2. Spatio-temporal evolution of EE

**4.2.1. Time evolution trend of EE.** Fig 5 illustrates that, on the whole, the mean EE of cities in the western region is mainly distributed between 0.9 and 1, showing a high level in general, but still not reaching DEA effectiveness. Regarding the tendency of time evolution, the overall average efficiency growth rate is 2.2%, showing a slight fluctuation upward trend. The highest value was 0.958 in 2017. Among them, the fluctuation range of the northwest region is small, showing a relatively stable trend as a whole. The southwest region has increased from 0.874 in 2007 to 0.913 in 2021, at a 4.3% growth rate, which indicates that the EE of the southwest region develops faster than that of the northwest region.

**4.2.2. Spatial evolution trend of EE.** In this part, we analyzed the measured results of urban EE in the western region, and used Arcgis 10.8 to carry out spatial visualization operations to study its spatio-temporal variation characteristics. To more clearly illustrate the spatial change properties of EE, it was split up into three distinct levels: low [0–0.8], medium [0.8–1] and high (>1), and provided a description of the urban EE spatial distribution map in western China between 2007 and 2021.

As seen by Fig 6, the overall performance of cities in western China is relatively high, with great regional differences. From 2007 to 2021, the quantity of low-level cities decreased by 8, the quantity of cities in the middle level increased by 11, and the quantity of cities in the high level decreased by 3, achieving a small improvement in general. Among them, the efficiency value of Chongqing was the lowest from 2007 to 2016, but it was at a high level in 2021, which may be due to the promulgation of Chongqing Environmental Protection Regulations, Chongqing Air Pollution Prevention and Control Regulations and other policies in 2017, which greatly contributed to the promotion of the growth of Chongqing's ecological environment.

From the perspective of spatial distribution, Western cities that abut the center of the region are home to a high concentration of EE, and the cities with low EE are mainly distributed in Guangxi Province and Guizhou Province. Consequently,

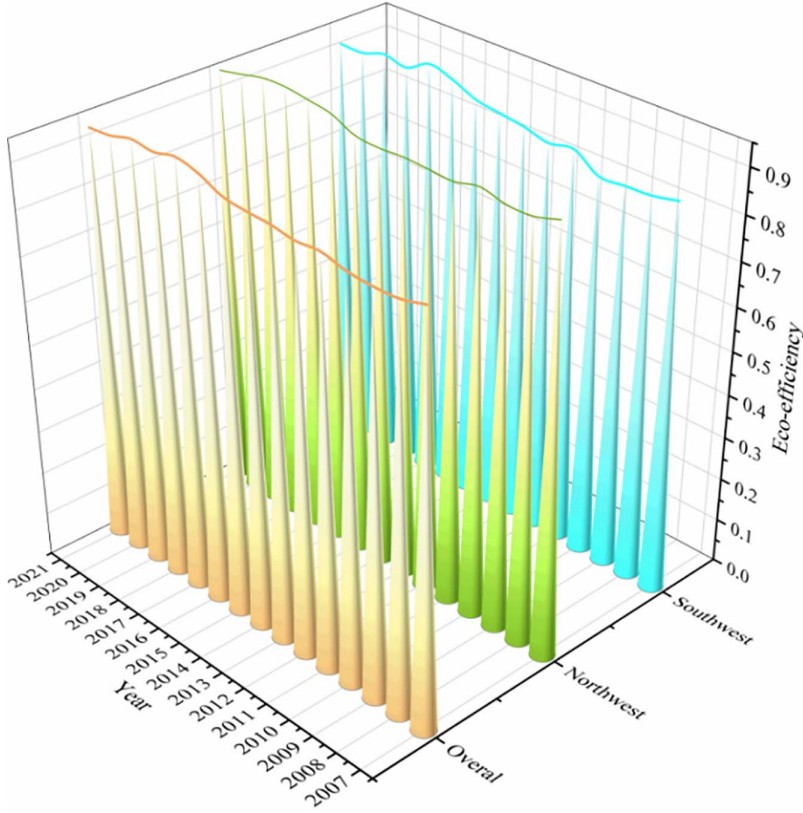

**Fig 5. Temporal evolution of urban EE in western China.**

consideration should be given to the sensible distribution of resources and technologies, and differentiated ecological protection policies should be formulated to achieve the improvement of the EE of the whole western region and each city.

### 4.3. Spatiotemporal variations of CCD-GFEE

**4.3.1. Time evolution of CCD-GFEE.** Fig 7 shows the time evolution trend of CCD-GFEE of cities in western China from 2007 to 2021. On the whole, CCD-GFEE in western China is mainly distributed between 0.4–0.5, and is in the near incoordination stage. From 0.407 in 2007 to 0.479 in 2021, the growth rate is 17.7%, the overall change is little, showing a small growth trend. The findings indicate that the CCD-GFEE of Western region cities has been improved in recent years, but the improvement is small, indicating that there is still space for the government to get better in this aspect. Among them, the CCD-GFEE of cities in northwest China increased from 0.414 in 2007 to 0.481 in 2021, at a pace of 16.2% growth; the CCD-GFEE of cities in southwest China increased from 0.401 in 2007 to 0.477 in 2021, with a 19.0% growth rate. This suggests that CCD-GFEE's growth rate in the southwest region is higher than that in the northwest region, which may be due to the fact that the southwest region attaches more importance to economic development, has a stronger awareness of environmental protection, a more reasonable allocation of resources and technology, and a more active financial market, so the CCD-GFEE has achieved faster development.

**4.3.2 Spatial evolution of CCD-GFEE.** To better study the CCD relationship between urban GF and EE in western China, this study divides the CCD into 10 levels according to the practice of Zhang, Geng [60], as shown in Table 3:

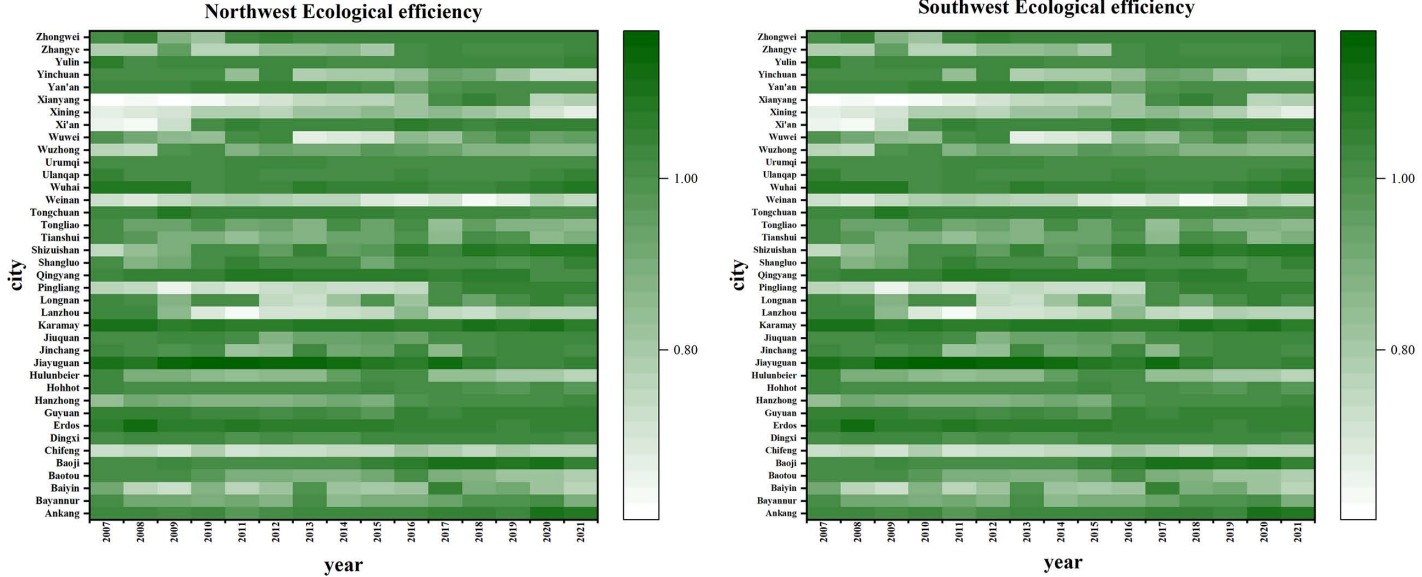

**Fig 6. Spatial evolution of urban EE in western China.**

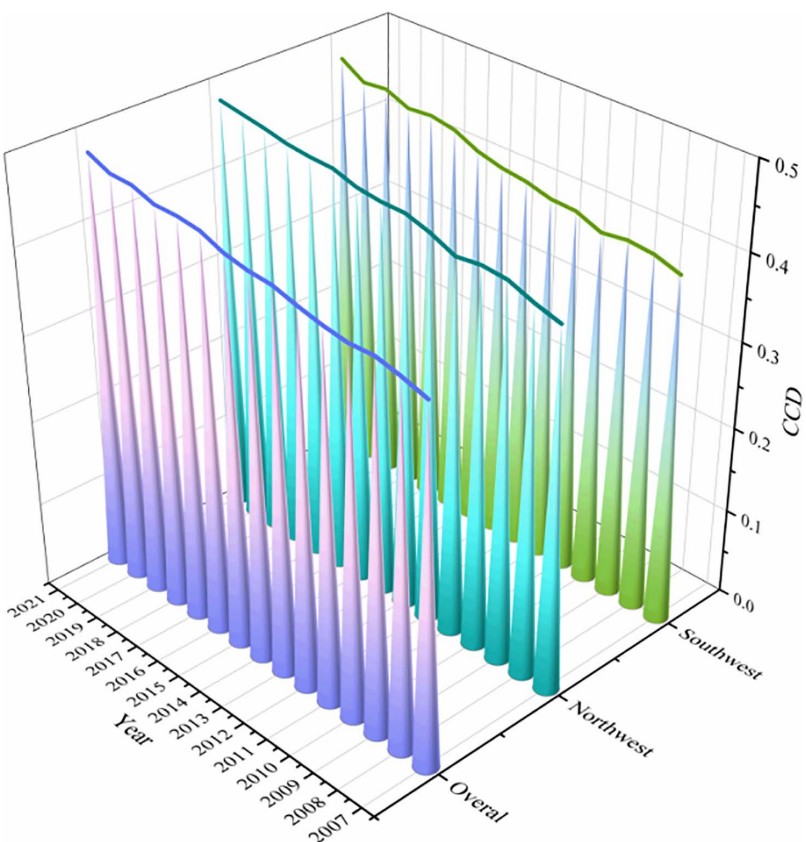

**Fig 7. Temporal evolution of CCD-CFEE in cities in the western region.**

Fig 8 shows the mean value of CCD-GFEE of cities in the western region. Cities with higher CCD-GFEE are primarily in the provinces of Shaanxi, Sichuan, and InnerMongolia. Among them, the CCD of Chengdu is the highest, reaching 0.592, which is in the Barely coordination stage.

**Table 3. CCD-GFEE evaluation criteria.**

| Coupling coordination level | Type | Coupling coordination level | Type |
|---|---|---|---|
| [0.000, 0.100) | Extreme incoordination | [0.500, 0.600) | Barely coordination |
| [0.100, 0.200) | Serious incoordination | [0.600, 0.700) | Primary coordination |
| [0.200, 0.300) | Moderate incoordination | [0.700, 0.800) | Intermediate coordination |
| [0.300, 0.400) | Mild incoordination | [0.800, 0.900) | Good coordination |
| [0.400, 0.500) | Near incoordination | [0.900, 1.000] | High-level coordination |

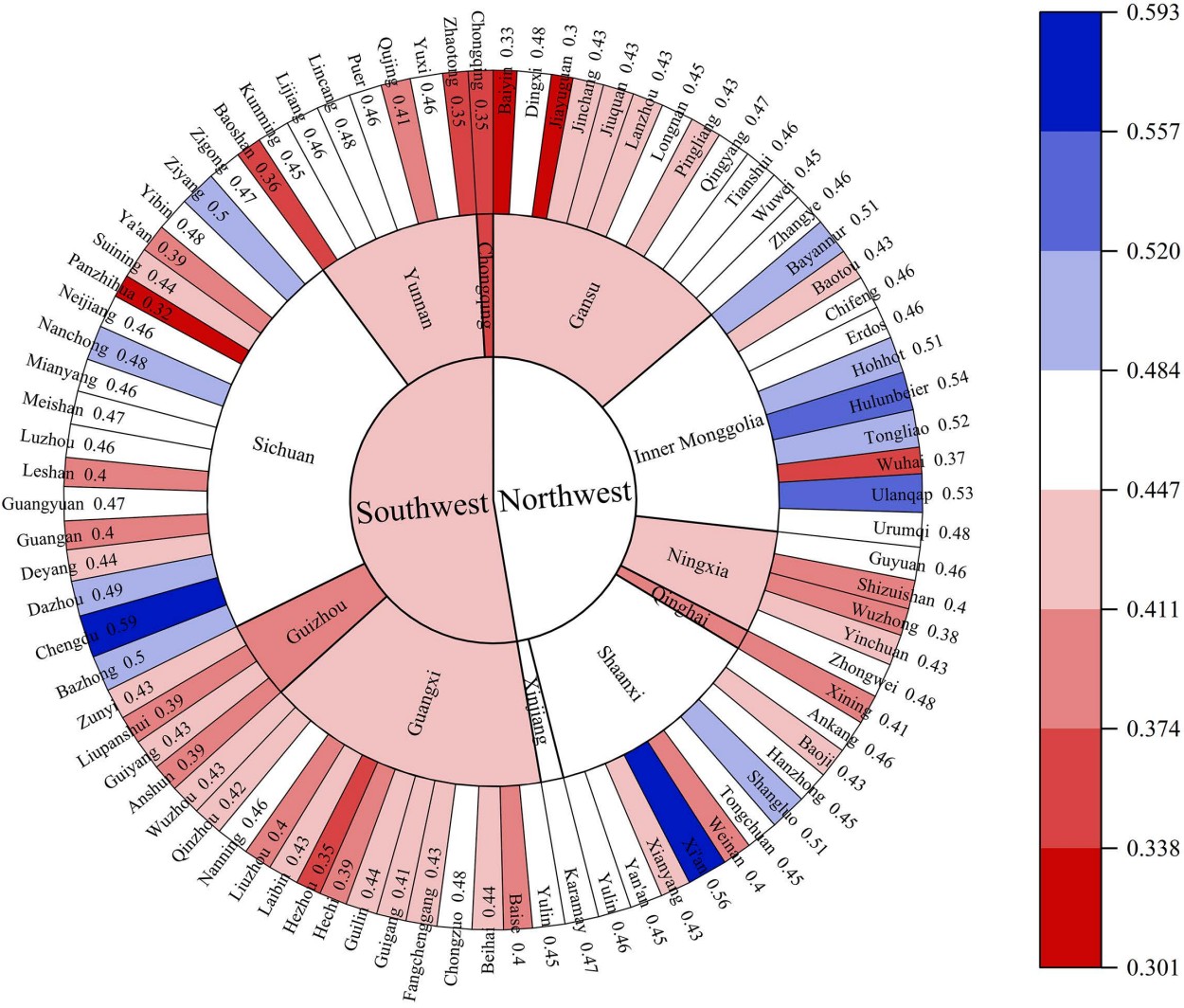

**Fig 8. Mean value of CCD-GFEE of cities in the western region.**

The regional distribution of CCD-GFEE in the western region is depicted in Fig 9 and Fig 10, It can be seen that the spatial distribution of CCD-GFEE presents signifiantly enhanced coupling coordination, but the CCD is still quite low. The evolution of GF and EE led to the expansion of the region where CCD-GFEE value increased, and regional differentiation was obvious. The CCD in northwest China is higher than that in Southwest China. During the period 2007–2021, CCD-GFEE's proportion of L3 cities is 9.5%,2.4%,1.2% and 0%, respectively. The proportion of cities in L4 was 31%, 32.1%, 11.9% and 13.1%, the proportion of cities in L5 was 51.2%, 54.8%, 72.6% and 52.4%, and the proportion of cities in L6 was 8.3%, 10.7%, 11.9% and 30.9%. In 2016, there will be two L7 cities, Xi 'an and Chengdu, and in 2021, there will be three L8 cities, Xi 'an, Chengdu and Chongqing. The above results show that the cities with low CCD-GFEE in western China are decreasing, while those with high CCD-GFEE are increasing, and the cities with the best performance have reached the L8 stage, that is, the Intermediate coordination stage. However, most cities are mainly distributed in the L5

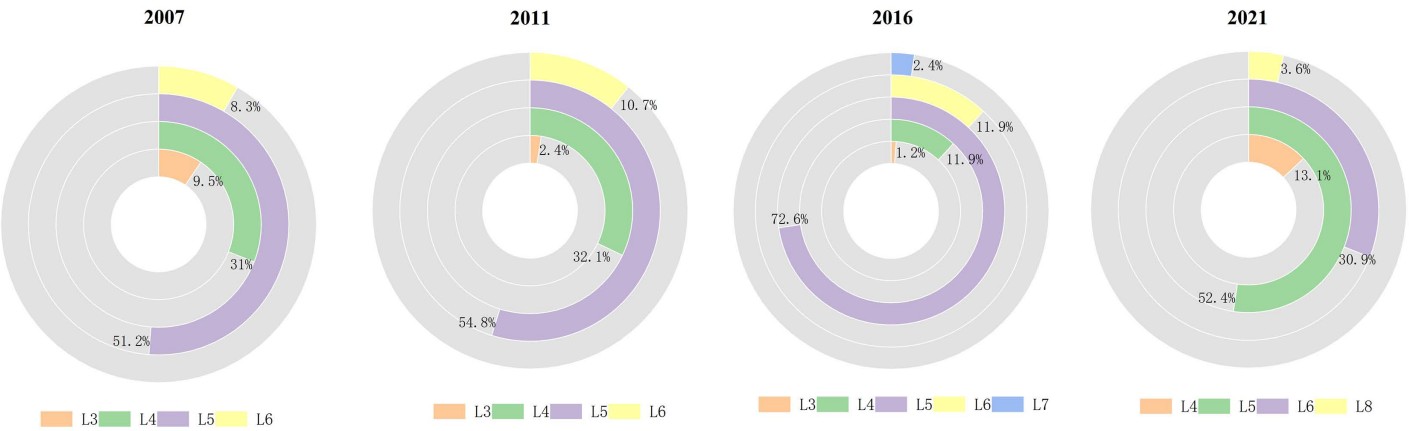

**Fig 9. Distribution of CCD-GFEE in cities in the western region.**

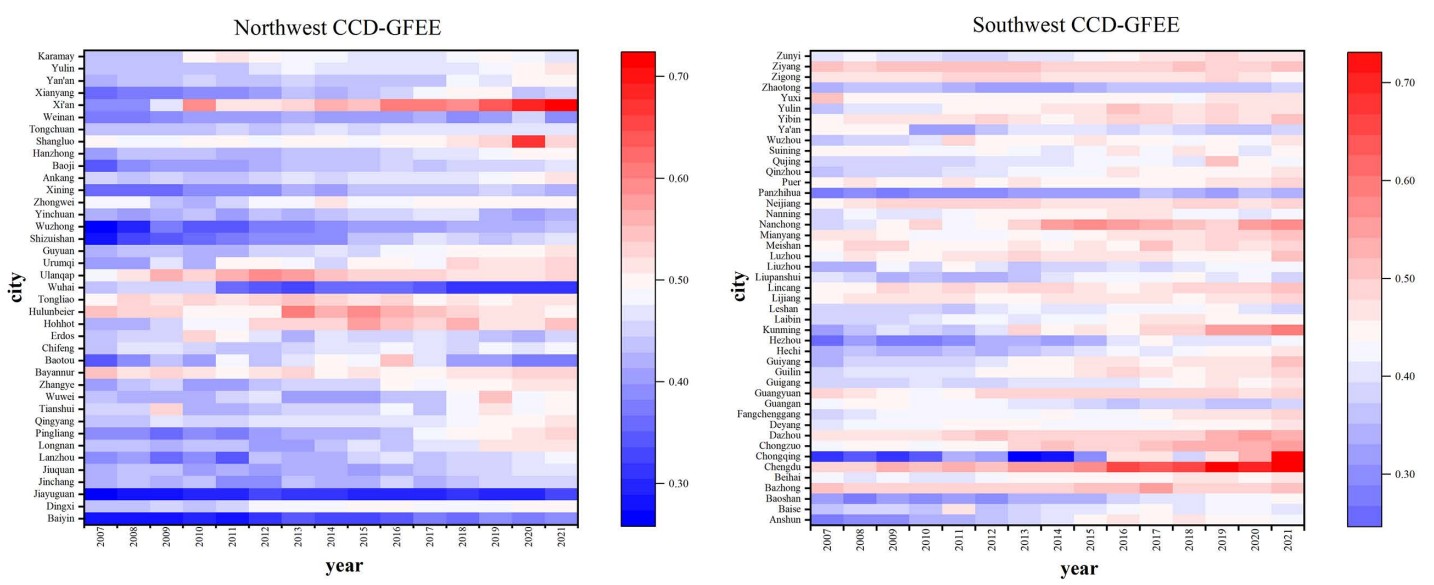

**Fig 10. Spatial distribution of CCD-GFEE in cities in the western region.**

level, that is, the Barely coordination stage. The CCD-GFEE of Xi 'an City, Chengdu City and Chongqing City has reached the Intermediate coordination stage. These cities are provincial capitals or national central cities, with a prime geographic location, advanced financial and economic development, the use of more advanced clean technologies, and better ecological environment quality, so their CCD-GFEE is higher. The cities with low CCD-GFEE are Baiyin, Panzhihua and Jiayuguan, all of which are rich in mineral resources and mainly rely on mineral resource development as their economic pillar. Due to resource consumption and pollution emission brought about by overdevelopment, their ecological efficiency is low, which leads to the inability to achieve coordinated growth between the natural environment and economic development. Among them, Chongqing has achieved the largest stage leap, transforming from Moderate incoordination in 2007 to Intermediate coordination stage in 2021. It shows that the policies and measures of GF and ecological environment in Chongqing have achieved obvious results at this stage.

**4.3.3. Spatial autocorrelation of CCD-GFEE.** To explore CCD-GFEE's spatial distribution properties in greater depth, the ArcGis10.8 software is used to calculate the global Moran's I of cities in western China, as shown in Table 4. The global Moran's I of CCD-GFEE passes the 5% significance level test in most years, and the worth of Moran's I exceeds 0, indicating that the spatial correlation is positive, showing the characteristics of spatial aggregation. We observe that the global Moran index and Z-value show a trend of initial increase followed by decline, indicating that the spatial aggregation effect of CCD-GFEE first increases and then decreases.

## 4.4. Analysis of CCD-GFEE's driving forces

**4.4.1. Model selection.** Based on the results of the global Moran index mentioned earlier (see Table 4), CCD-GFEE has significant spatial correlation in space. That is the coupling coordination degree and its influencing factors do not exist independently in space but are spatially correlated. Moreover, the research data are urban panel data with spatial attributes. Traditional methods may lead to result bias due to ignoring spatial effects. The spatial econometric model can capture spatial correlations and correct biases, and thus is more suitable for measuring the influencing factors of CCD-GFEE. In this research, the conventional method of least squares is first used to conduct preliminary simulation of the model, and on this basis, the spatial autocorrelation model's residual terms is tested by Matlab7.12 software, in order to confirm that the spatial model is appropriate. Table 5 displays the individual regression results. In addition, to show the necessity of the model to control the fixed effect. Additionally, Table 5 provides the results of the regression of four distinct model types, such as non-fixed effect, time fixed effect, spatial fixed effect and two-way fixed effect. By comparing the four models, it can be determined which fixed effect model has the strongest explanatory power.

Based on the findings presented in Table 5, the judgment coefficients of goodness of fit of a comparison is made between the four models. The R-squared values of the no-fixed effect model, the temporal fixed effect model, the spatial fixed effect model, and the bidirectional fixed effect model are 1.7829, 0.0748, 0.31886 and 0.0663, respectively. It is evident that the best performing model is the spatial fixed effect model. It demonstrates that out of the three models, the spatial fixed effect model fits data better. Consequently, contrary to the other three models, the spatial fixed effect model has the best explanatory power, so this paper finally chooses this model for the subsequent empirical analysis. Simultaneously, the second portion of Table 1 investigates the presence of spatial self-phase in the model's residual terms. According to the findings, LM-lag is 9.2958 in the spatial fixed effect model, passing the 1% significance level test. LM-err was

**Table 4. Moran's I value, Z value and P value of CCD-GFEE.**

| Year | Moran's I | Z-value | P-value |
|------|-----------|---------|---------|
| 2006 | 0.1176 | 2.8140 | 0.0049 |
| 2010 | 0.1469 | 3.4354 | 0.0006 |
| 2016 | 0.0914 | 2.2721 | 0.0231 |
| 2021 | 0.0025 | 0.3248 | 0.7453 |

**Table 5. Estimate and test results of common panel data model.**

| variable | No fixed effect | Space solid Fixed effect | Time fixation Fixed effect | Bidirectional solidification Fixed effect |
|---|---|---|---|---|
| ER | 0.0337*** (3.1692) | 0.0056 (0.7195) | 0.0281** (2.4517) | 0.0115 (1.4569) |
| PD | 0.0060*** (2.6388) | −0.0084*** (−3.7538) | 0.0063*** (2.7898) | −0.0079*** (−3.5798) |
| IS | 0.0762*** (4.1323) | 0.0845*** (4.8880) | 0.0475** (2.3297) | 0.0213 (0.8095) |
| EL | 0.0087*** (2.5814) | 0.0264*** (6.5127) | 0.0046 (1.1207) | 0.0300*** (4.4229) |
| TP | 0.0101*** (5.4080) | 0.0105*** (4.6837) | 0.0111*** (5.7470) | 0.0124*** (5.2356) |
| GI | 0.05721*** (3.5524) | −0.0326* (−1.7484) | 0.0509*** (2.6908) | −0.0024 (−0.1117) |
| $R-squared$ | 0.1645 | 0.3188 | 0.0748 | 0.0663 |
| $DW$ | 1.7829 | 1.6182 | 1.7880 | 1.6457 |
| $LM-lag$ | 64.1679*** | 9.2958*** | 45.4330*** | 3.1570* |
| $Robust\ LM-lag$ | 72.3860*** | 0.3814 | 19.7228*** | 1.1350 |
| $LM-err$ | 40.6863*** | 8.9854*** | 36.0139*** | 2.4427 |
| $Robust\ LM-err$ | 48.90431*** | 0.0710 | 10.3036 *** | 0.4207 |

Note: () data are T-test values, the symbols *, **, and *** denote significance levels of 10%, 5%, and 1%, in that order;The model estimate and spatial autocorrelation test were performed using Matlab 7.12.

8.9854, which was also significant at 1% level. These two findings confirm the existence of spatial autocorrelation in the model's residual term, and the traditional least square estimation cannot solve this problem, so the traditional model's estimation results could be skewed. Thus, it is necessary to transform the ordinary model into a spatial model. Furthermore, because LM-lag's statistics are greater than LM-err's, the SAM is more suitable for this paper than the SEM.

**4.4.2. Estimation results of spatial econometric model.** There will unavoidably be bias because the spatial autocorrelation issue of the residual term cannot be resolved by the estimation results of the ordinary model. Thus, the spatial model is re-simulated in this article using the maximum likelihood method, yielding SAR and SEM estimate results, respectively. Table 6 displays the outcomes, both $W*dep.$ var. of the SAR and $spat.\ aut.$ of the SEM passed a 1% significance level on the test, which once again verified the suitability of the use of the spatial model. In comparison with the standard model's regression findings, on the one hand, the decision coefficient $R-squared$ of the goodness of fit of the spatial model increases, indicating that the explanatory power of the model is further enhanced. On the other hand, the spatial model's variable regression coefficients agree with the ordinary model's values, but the T-test of some variables is improved, which reflects that the estimation's outcome of the spatial econometric model are optimized on the basis of the common model. Besides, the statistics of LM-lag mentioned above are larger than those of LM-err. In comparison, the SAR has a stronger explanatory power than the SEM. Therefore, this paper chooses the SAR's estimation results to carry out the meaning interpretation of the variable results.

As Table 6 illustrates, that ER has no significant impact on CCD-GFEE. This may be because although environmental regulations can greatly improve the ecological environment, the economic consumption brought by them cannot be ignored, so the impact on CCD-GFEE is not significant.

PS affects CCD-GFEE negatively at the 1% significant level. This indicates that population growth is not conducive to the improvement of CCD-GFEE, it might be because the rate of urban population increase may momentarily outpace the

**Table 6. Estimate and test results of spatial econometric model (spatial fixed effect model).**

| Variable | SAR | SEM |
|---|---|---|
| ER | 0.0071<br>(0.9113) | 0.0058<br>(0.7458) |
| PS | −0.0080***<br>(−3.5781) | −0.0088***<br>(−3.9466) |
| IS | 0.1095***<br>(6.0375) | 0.0713***<br>(3.7201) |
| EL | 0.0335***<br>(7.3760) | 0.0271***<br>(6.3141) |
| TP | 0.0112***<br>(4.8925) | 0.0109***<br>(4.7640) |
| GI | −0.0325*<br>(−1.7271) | −0.0271<br>(−1.3965) |
| $W * dep.$ var. | -0.2360***<br>(−3.9009) | |
| $spat. aut.$ | | 0.2070***<br>(3.6817) |
| $R - squared$ | 0.7378 | 0.7442 |

Note: () data are T-test values, *, **, *** indicate significance levels of 10%, 5% and 1% respectively.

rate of economic growth and the environmental resources' carrying capacity. that accompany urbanization, resulting in a serious waste of resources [60], which is not advantageous for the coordinated growth of GF and environment.

The effect of IS on CCD-GFEE is positive at 1% significance level. It demonstrates that rising tertiary industry proportions are favorable to rising CCD-GFEE. The reason may be that, in contrast to secondary industry, the tertiary sector that is service-based consumes less resources and produces less air pollution and solid waste [60].

The effect of EL on CCD-GFEE is positive at 1% significance level. Economic development can effectively support construction of infrastructure, scientific research, and environmental protection [60], to achieve the GF's coordinated growth with the ecological environment.

The effect of TP on CCD-GFEE is positive at 1% significance level. Environmentally friendly product development can be aided by technological innovation [78], Thus, it will stimulate greener corporate financing activities and promote the growth of GF. In addition, new environmentally friendly technologies driven by scientific and technological progress have reduced pollution emissions and improved resource utilization efficiency [79].

The effect of GI on CCD-GFEE is positive at 10% significance level. Misallocation of resources caused by government intervention may induce over-investment and overcapacity of enterprises. When the support for resource-based enterprises is greater than that for green enterprises, it is not helpful for the ecological environment's and GF's coordinated growth.

## 5. Conclusion and discussion

(1) On the whole, green finance generally shows a low level. This might be due to the fact that the number of financial institutions in the western region is small and their distribution is sparse, and the local fiscal and enterprise financial strength is limited, making it difficult for them to bear the high initial investment and long payback period of green projects. In addition, the western region is an ecologically fragile area in China. However, it has long relied on resource-based industries to drive its economy. The green transformation of industries requires high costs. In the short term, it is more inclined to maintain the stability of traditional industries, resulting in insufficient policy implementation

and resource input for GF support. From the standpoint of the evolution of time, The GF level primarily exhibits an increased trend between 2007 and 2021, which agrees with the research done by Yi, Hao [80], and the southwest region is growing more quickly than the northwest region. From the standpoint of the evolution of space, the overall development level of urban GF in western China is still rather low, with large regional differences, the study of Yuan, Jia [81] also shows that GF shows obvious regional differences. In addition to the high level of GF development in some cities, the level of GF in other regions is always low, which indicates that most cities in western China have plenty of space for improvement. In space, The northwest area of the high level city has gradually given way to the southwest area. The southwest region enjoys the support of national strategies such as the Chengdu-Chongqing Twin-City Economic Circle and Kunming as a regional international center city, making its policy dividends more concentrated. The overall economy in the northwest region is fragmented and regional coordination is insufficient, making it difficult to promote the development of GF. In addition, the industrial structure in the southwest region is greener, while that in the northwest region relies on high-carbon industries and has a prominent feature of heavy industrial structure.

(2) On the whole, the urban ecological efficiency shows a relatively high level, but it is still not DEA effective. As shown by Xue, Yue [5], the EE in the western region is low. This may be due to the fact that the public service input is included in the index system in this paper, which leads to the difference while determining the efficiency value. From the standpoint of the evolution of time, the overall average efficiency shows a slightly fluctuating upward trend, This is also confirmed by the research of [82]. The highest value of ecological efficiency appeared in 2017, however, the urban ecological development in the southwest region is faster than that in the northwest region. which agrees with the research done by [5]. Northwest China leads in efficiency values thanks to the short-term efficiency advantage of resource-based industries, but ecological fragility and transformation pressure have limited its development speed. Although the short-term efficiency in Southwest China is relatively low due to high ecological investment, the synergy of natural conditions, policy support and industrial transformation has accelerated the pace of ecological development. From a spatial standpoint, the western region of China has a higher level of urban performance as a whole, and the variations amongst regions are substantial. Southwest area efficiency is lower than northwest region efficiency. which agrees with the research done by Xue, Yue [5], The cities with low EE are mainly located in Guangxi, Guizhou and Gansu provinces. This is similar to the study of Xue, Yue [5]. The spatial difference of EE between cities in Guangxi Province and Gansu Province is large, which indicates that the EE of cities in Guangxi and Gansu province is not balanced.

(3) From the standpoint of the evolution of time, the CCD-GFEE of cities in western China is mainly distributed between 0.4–0.5, with little overall change and a slight increase trend. This might be due to the lagging development of GF, which has limited support capacity for EE. Meanwhile, the "low starting point and slow improvement" of EE in the western regions have restricted the coordinated development with GF. This is different from the research of Zhang [83], This may be due to the fact that the study's scope is different from the prefecture-level city in this study. From the standpoint of the evolution of time, the CCD-GFEE's spatial correlation is positive, showing obvious aggregation effect, which agrees with the research done by Zhang, Geng [60], This might be due to the lagging development of GF, which has limited support capacity for EE. Meanwhile, the "low starting point and slow improvement" of EE in the western regions have restricted the coordinated development with GF. The spatial aggregation effect of CCD-GFEE first went up, then down. This might be because as the development stage advanced, the internal differentiation within the region intensified, breaking the initial synchronicity. Significantly improved coupling coordination was evident in the western region's CCD-GFEE spatial distribution, but the CCD was still at a low level, and the regional differentiation was obvious, Zhang, Geng [60] came to a similar conclusion. This might be due to the disparity in resources, policies and industries between core cities and medium and small-sized cities, which has exacerbated regional differentiation. Inner Mongolia and the provinces of Shaanxi and Sichuan are home to the majority of the cities with high CCD-GFEE.

The CCD of Chengdu city is the highest, which is in the stage of almost coordination. The CCD in northwest China is higher than that in Southwest China, this might be due to the fact that the resource-based industries in the northwest region initiated a systematic green transformation earlier. Moreover, relying on its advantages in wind and solar energy resources, the northwest region has formed a complete new energy industrial chain. Additionally, the governance of ecologically fragile areas in the northwest region is more urgent, and policy and financial input are more precise. The cities with low CCD-GFEE in western China are decreasing, while the cities with high CCD-GFEE are increasing. Most cities are mainly distributed in the L5 level, that is, the Barely coordination stage. The CCD-GFEE of Xi 'an, Chengdu and Chongqing have reached the Intermediate coordination stage, and all of these cities serve as either national central capitals or provincial capitals. The cities where the CCD-GFEE is always low are Baiyin, Panzhihua and Jiayuguan, and Chongqing has achieved the largest phased leap-over.

(4) Viewed through the lens of CCD-GFEE driving factors, environmental regulations' effects on CCD-GFEE is not significant, the economic development level in the western region is relatively lagging behind. To ensure employment and short-term economic growth, local governments may have a tendency towards "lenient" environmental regulations. The constraints imposed by environmental regulations on enterprises' polluting behaviors are insufficient, making it difficult to force enterprises to enhance EE through green technological innovation. Meanwhile, due to the lack of clear regulatory guidance and incentives, green financial tools are also difficult to form synergy with environmental regulations, resulting in an insignificant promoting effect on the degree of coupling and coordination. While the impact of population size and government intervention on CCD-GFEE is negative, The ecological environment in the western region is fragile and its environmental carrying capacity is limited. The expansion of population size will directly increase resource consumption and pollution emissions, leading to a decline in EE. The results of Zhang, Geng [60] are close to this. However, different from the research conclusion of Zhang [83], This may be due to different government industrial policies in different regions. The influence of industrial structure, economic level and technological progress on CCD-GFEE is positive. The optimization of industrial structure is often accompanied by the agglomeration of green industries. Such agglomeration can not only reduce the ecological cost per unit output through the scale effect, but also attract the concentrated investment of green financial resources, there by increasing the CCD-GFEE. The improvement of economic level lays the foundation for the coordinated development of GF and EE by enhancing the capacity of resource supply and optimizing development goals. The positive impact of technological progress on the degree of coupling coordination is reflected in not only directly enhancing EE but also providing application scenarios and risk control tools for GF, achieving a deep integration of the two, Zhang, Geng [60] came to a similar conclusion.

## 6. Policy suggestions and prospects

### 6.1. Policy suggestions

On the whole, the urban GF and ecological environment system in western China have not achieved a highly coordinated development. The work proposes the following pertinent policy recommendations to encourage the coordinated growth of GF and EE in different regions.

(1) China's GF standard system should be clarified. In terms of the national level, An integrated GF framework should be formulated, and the percentage of the system's environmental indicators should be appropriately increased. Secondly, we should actively expand the scope of GF, constantly explore new green financial instruments, and enrich the catalogue of green financial instruments. Furthermore, there should be an expansion of the green financial market, and governments at all levels should actively attract social capital into green projects, such as green transportation, innovative energy, and clean energy. Finally, local governments develop green finance according to local actual conditions, so that funds can flow vigorously to green projects.

(2) Strengthening economic development. Strengthening connection and communication among cities, encouraging the various elements to flow freely is essential. Furthermore, it is necessary to rationally allocate the resource elements among cities, create an ecological compensating mechanism that spans regions, optimize the allocation of resources, capitalize on each region's special characteristics and benefits and realize the coordinated economic development of each region.

(3) Promoting the modification of the industrial structure. The resource-mining and resource-processing industries in the western region are important economic pillars. Enterprises should promote their own green transformation and achieve the shift in the industrial sector from "terminal governance" to "clean production." For resource-based cities, industrial transformation should be promoted, and industries with high consumption and high pollution should be shifted to industries with low consumption and low pollution, in order to encourage the green development of industries. We should increase the percentage of the tertiary sector, and at the same time, vigorously develop energy-saving and environment-friendly enterprises, optimize the energy structure, and increase the effectiveness of resource use.

(4) Improve the technical level. The government should strengthen the training of technical personnel and attract more talents to participate in the technical field. Strengthen the guidance and support of local governments, increase the support for technological innovation, encourage businesses to spend more on technological research and development and improve the local technology level.

(5) Rationally regulate industrial policies. The government should optimize the green allocation of finance, optimize the green finance allocation based on national finance, and increase capital investment in green infrastructure. In terms of green finance, the state should reduce a certain tax proportion and introduce relevant subsidy policies in terms of scientific and technological innovation. Secondly, relevant government departments can take a series of measures such as weakening the risk weight of green assets, taking targeted reduction of reserve requirement ratio in terms of credit, and taking priority compensation in terms of bonds. Introduce more financial institutions and the active participation of the public to ensure more funds for the healthy development of green finance.

## 6.2. Limitations and prospects

This study has enriched the empirical study of CCD-GFEE at the prefecture level, analyzed its temporal and spatial evolution characteristics, and explored its driving factors by using a spatial econometric model. However, there are still certain restrictions on this study, which point to the path of further investigation. First of all, limited by the availability of data, it is difficult for the current indicator system to fully depict the full connotation of urban green finance, and more research is required to develop a more sensible and thorough urban green finance measurement indicator system. Secondly, the purpose of this paper's research is prefecture-level cities and above. If further study on the CCD of the two systems is carried out by county-level cities, the regional differences, features of the CCD-GFEE's spatiotemporal evolution and driving variables will be more deeply revealed, which will have greater application value for local governments at all levels to implement sustainable development strategies. Finally, the field of study covered in this work is the western region of China, which can be further expanded, and more detailed classification of types of cities can be carried out according to different resource endowment and industrial structure characteristics, which is also one of the contents of the follow-up research of this paper.

## Author contributions

**Conceptualization:** Dalai Ma.

**Data curation:** Bitan An, Ruonan Chang.

**Formal analysis:** Jiawei Zhang.

**Funding acquisition:** Fengtai Zhang.

**Investigation:** Zuman Guo.

**Methodology:** Zuman Guo.

**Resources:** Fengtai Zhang.

**Software:** Dalai Ma.

**Supervision:** Yin Yan.

**Visualization:** Jiawei Zhang.

**Writing – original draft:** Dalai Ma.

**Writing – review & editing:** Bitan An.

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
