## [Decision Letter · Decision Letter 0]

21 May 2025

PONE-D-24-45220What are the spatio-temporal differentiation characteristics and driving factors of the coupling coordination degree between green finance and ecological efficiency? Evidence from 84 cities in western ChinaPLOS ONE?

Dear Dr. An,

Thank you for submitting your manuscript to PLOS ONE. After careful consideration, we feel that it has merit but does not fully meet PLOS ONE’s publication criteria as it currently stands. Therefore, we invite you to submit a revised version of the manuscript that addresses the points raised during the review process.

We look forward to receiving your revised manuscript.

Kind regards,

Xiaoyong Zhou, Ph.D.

Academic Editor

PLOS ONE

Journal Requirements:

4. We note that Figures 1, 5-1, 5-2, 5-3, 5-4, 7-1, 7-2, 7-3, 7-4, 10-1, 10-2, 10-3 and 10-4 in your submission contain [map/satellite] images which may be copyrighted. All PLOS content is published under the Creative Commons Attribution License (CC BY 4.0), which means that the manuscript, images, and Supporting Information files will be freely available online, and any third party is permitted to access, download, copy, distribute, and use these materials in any way, even commercially, with proper attribution. For these reasons, we cannot publish previously copyrighted maps or satellite images created using proprietary data, such as Google software (Google Maps, Street View, and Earth). For more information, see our copyright guidelines: http://journals.plos.org/plosone/s/licenses-and-copyright .

     1. You may seek permission from the original copyright holder of Figures 1, 5-1, 5-2, 5-3, 5-4, 7-1, 7-2, 7-3, 7-4, 10-1, 10-2, 10-3 and 10-4 to publish the content specifically under the CC BY 4.0 license. 

Reviewers' comments:

Reviewer's Responses to Questions

**Comments to the Author**

1. Is the manuscript technically sound, and do the data support the conclusions?

Reviewer #1: Yes

Reviewer #2: Yes

2. Has the statistical analysis been performed appropriately and rigorously?

Reviewer #1: Yes

Reviewer #2: Yes

3. Have the authors made all data underlying the findings in their manuscript fully available?

Reviewer #1: Yes

Reviewer #2: No

4. Is the manuscript presented in an intelligible fashion and written in standard English?

Reviewer #1: Yes

Reviewer #2: No

Reviewer #1: This paper empirically analyzes the spatio-temporal characteristics and driving factors of the coupling coordination degree between green finance and ecological efficiency by using panel data encompassing 84 cities in Western China spanning from 2007 to 2021. There are a few suggestions for further revision:

1.It is suggest to write questions at the end of introduction, and write what methods/approaches you will use to answers of those study questions.

2.The fourth point of innovation in the paper does not seem to be an innovation point, but rather a research significance.

3.What are the selection principles for the 84 cities in the research sample?

4.Many formulas, letters in the paper are not neat and need to be standardized.。

5.The clarity of the images in the text is low and needs to be modified.

6.The font formats in Table 1 are not consistent.

7.Table 6 does not use a three line table and is not consistent with the previous table format.

8.Why did the spatial econometric model be selected to measure the influencing factors of coupling coordination degree, rather than other methods? This is not clearly stated in the paper.

9.The citation format for references is inconsistent, with some citing the first letter of the name and others not.

Reviewer #2: Based on panel data from 84 cities in western China spanning 2007 to 2021, this paper constructs a comprehensive green finance (GF) indicator system and explores the coupling coordination degree (CCD) relationship between GF and eco-efficiency in western Chinese cities at the municipal level, thereby expanding our understanding of the interaction between GF and EE. However, several issues require revision:

(1) The logic in section 3.3.1 lacks clarity. The steps of the entropy weight method are not sequentially labeled from the first step. Additionally, there are two sections numbered 3.1, which needs correction.

(2) The paper’s readability could be improved. It is recommended to insert tables and figures within the text to enhance readability and facilitate review. Furthermore, the current images are blurry and need to be replaced with higher-quality versions.

(3) On line 495, the meaning of “NOTE” is unclear, and this annotation is not properly identified within the text.

(4) Some section titles do not accurately reflect their content. For instance, in section 3.4.3, the title “Driver index construction of CCD-GFEE” does not match the content, which states, “This section adopts spatial econometric modeling and other methods to further explore the driving elements behind the coordinated development of the two.”

(5) The analysis of calculation results lacks depth. It primarily focuses on surface-level data interpretation without delving into specific underlying causes.

**Do you want your identity to be public for this peer review?** For information about this choice, including consent withdrawal, please see our Privacy Policy

Reviewer #1: **Yes: ** Jingjie Li

Reviewer #2: **Yes: ** Jiexin Tang

---

## [Author Response · Author response to Decision Letter 1]

17 Aug 2025

Response to Reviewers

Dear Editor,

Thank you for arranging the review and the reviewer's valuable comments. We have read your manuscript in detail and submitted the Response to Reviewers and the Revised Manuscript with Track Changes and the Manuscript. In addition, since we have modified the pictures in the original text, we have uploaded a new file called "Modified Pictures". According to the two reviewers' comments. All the changes have been marked. Thanks again to the editors and reviewers for their help.The following are the responses to the editor and reviewers:

---

## [Decision Letter · Decision Letter 1]

22 Sep 2025

What are the spatio-temporal differentiation characteristics and driving factors of the coupling coordination degree between green finance and ecological efficiency? Evidence from 84 cities in western China

PONE-D-24-45220R1

Dear Dr. An,

We’re pleased to inform you that your manuscript has been judged scientifically suitable for publication and will be formally accepted for publication once it meets all outstanding technical requirements.

Kind regards,

Xiaoyong Zhou, Ph.D.

Academic Editor

PLOS ONE

Additional Editor Comments (optional):

Reviewer #1:

Reviewers' comments:

Reviewer's Responses to Questions

**Comments to the Author**

Reviewer #1: All comments have been addressed

2. Is the manuscript technically sound, and do the data support the conclusions?

Reviewer #1: Yes

3. Has the statistical analysis been performed appropriately and rigorously?

Reviewer #1: Yes

4. Have the authors made all data underlying the findings in their manuscript fully available?

Reviewer #1: Yes

5. Is the manuscript presented in an intelligible fashion and written in standard English?

Reviewer #1: Yes

Reviewer #1: The authors have made detailed revisions to the paper based on the review comments, and I am satisfied with the current revised version of the paper.

**Do you want your identity to be public for this peer review?** For information about this choice, including consent withdrawal, please see our Privacy Policy

Reviewer #1: **Yes: ** Jingjie Li

---

## [Editor Report · Acceptance letter]

PONE-D-24-45220R1

PLOS ONE

Dear Dr. An,

I'm pleased to inform you that your manuscript has been deemed suitable for publication in PLOS ONE. Congratulations! Your manuscript is now being handed over to our production team.

Kind regards,

on behalf of

Dr. Prof. Xiaoyong Zhou

Academic Editor

PLOS ONE